# Echocardiographic Changes in Dogs with Stage B2 Myxomatous Mitral Valve Disease Treated with Pimobendan Monotherapy

**DOI:** 10.3390/vetsci11120594

**Published:** 2024-11-25

**Authors:** Andrew Crosland, Pablo Manuel Cortes-Sanchez, Siddharth Sudunagunta, Jonathan Bouvard, Elizabeth Bode, Geoff Culshaw, Joanna Dukes-McEwan

**Affiliations:** 1Department of Small Animal Clinical Science, School of Veterinary Science, University of Liverpool, Cardiology Service, Small Animal Teaching Hospital, Chester High Road, Neston CH64 7TE, UK; pcortescardio@gmail.com (P.M.C.-S.); sid.sudunagunta@veterinarycentre.co.uk (S.S.); lizbode@chestergates.org.uk (E.B.); 2The Royal (Dick) School of Veterinary Studies, Easter Bush Campus, The University of Edinburgh, Cardiology Service, Hospital for Small Animals, Midlothian EH25 9RG, UKgeoff.culshaw@ed.ac.uk (G.C.); 3Peripatetic Cardiology, Casar de Periedo, Cabezón de la Sal, 39591 Cantabria, Spain; 4Willows Veterinary Centre and Referral Service, Highlands Road, Shirley, Solihull B90 4NH, UK; 5ChesterGates Veterinary Specialists, Units E & F, Telford Court, Gates Lane, Chester CH1 6LT, UK

**Keywords:** canine, degenerative mitral valve disease, pimobendan, reverse remodelling

## Abstract

The present study aimed to evaluate the effects of pimobendan on heart size in dogs diagnosed with degenerative heart valve disease and left-sided heart chamber enlargement using ultrasound. Changes in left atrial and left ventricular size were compared over time between the treatment group and the control group (receiving no medication). Dogs receiving pimobendan showed a significant reduction in left ventricular size at the first follow-up compared to the control group. The reduction in heart size is termed reverse remodelling. Due to its documented effect of cardiac reverse remodelling, restraint is advised when prescribing pimobendan based on the detection of a heart murmur in patients where an initial heart ultrasound is an option. Some treatment dogs in this study no longer met the ultrasound criteria for left atrial and/or left ventricular enlargement, and their disease could have been misclassified with their medication inappropriately withdrawn. We suggest these dogs are referred to as reverse remodelled.

## 1. Introduction

Degenerative mitral valve disease, also known as myxomatous mitral valve disease (MMVD), is the most common acquired heart disease in dogs [1,2]. It is primarily diagnosed in older small to medium-sized breeds; however, large-breed dogs can also be affected. Breeds overrepresented with the disease include cavalier King Charles spaniels and dachshunds [1]. The auscultation of a left apical systolic heart murmur is typically the first indicator of the underlying pathology. The progression of the disease is characterised by volume overload of the left atrium and left ventricle, which may eventually lead to the development of left-sided congestive heart failure (CHF). Dogs with MMVD are currently classified according to the American College of Veterinary Internal Medicine (ACVIM) consensus statement [3] into stages A, B1, B2, C, and D. Stage B (both B1 and B2) includes dogs with an audible left apical systolic heart murmur prior to the onset of CHF. In stage B1, the left heart chambers remain within reference values or are minimally increased in size, so they do not satisfy the echocardiographic or radiographic criteria set by a large-scale clinical trial, the EPIC study [4]. In stage B2, both the left atrium and the left ventricle are dilated (remodelled) and satisfy the criteria set in the EPIC study [4]: a short axis left atrium to the aortic ratio in early diastole (LA/Ao) ≥ 1.6 and a M-mode-derived left ventricular internal diameter during diastole normalised for body weight by allometric scaling (LVIDdN) ≥ 1.7 [5].

Management of dogs with preclinical MMVD has evolved in recent years based on the evidence delivered by the EPIC study [4], which demonstrated that the administration of pimobendan to patients in stage B2 delayed the onset of CHF by a median of 15 months. The exact mechanism for this is not completely understood. No clinical trial has been conducted to determine whether pimobendan might be beneficial to stage B1 patients, and some authors have suggested deleterious effects in this population [6,7]. Thus, treatment with pimobendan is currently not indicated for this stage of MMVD [3].

Previous studies have assessed the effect of pimobendan in dogs with mitral regurgitation and cardiomegaly. In a population of dogs with MMVD and CHF (stages C and D), a reduction in the size of both the left atrium and left ventricle following pimobendan administration was described [8,9]. In another cohort of dogs with MMVD, a reduction in vertebral heart score (VHS) in dogs in CHF treated with pimobendan at one and three months post-initiation of treatment was observed [10]. In dogs with experimentally induced mitral regurgitation and cardiomegaly, a reduction in the left heart chamber sizes two and four weeks after starting treatment with pimobendan was observed [11]. In addition, the longitudinal data analysis derived from the EPIC study [12] identified a reduction in both the size of the left atrium and left ventricle at follow-up assessment 35 days after starting pimobendan treatment, but data post 35 days are lacking. A reduction in left ventricular size following pimobendan therapy was also documented in Dobermans with dilated cardiomyopathy [13]. This phenomenon, whereby the cardiac chambers show a decrease in size in response to treatment, is commonly termed “reverse remodelling”. As a result of increased awareness of the beneficial impact of pimobendan on dogs with MMVD, as well as new market authorisations for its use, the prescription of pimobendan has increased across the profession [14]. This, combined with reverse remodelling effects, has implications for MMVD staging, particularly when pimobendan is prescribed prior to cardiac imaging.

This study aimed to assess the effects of chronic pimobendan monotherapy on cardiac size in dogs diagnosed with stage B2 MMVD. In particular, we aimed to assess the impact of pimobendan on cardiac chamber sizes if prescribed prior to staging the MMVD.

## 2. Materials and Methods

This was a retrospective observational study. The authors obtained ethical approval from the ethics committees of two universities (VREC 600, VERC 36.18) in the UK. Owners provided informed consent for echocardiography and other clinically indicated procedures at the time of presentation. Owners also permitted use of their pet’s anonymised clinical data for research purposes.

### 2.1. Animals

Client-owned dogs that were presented to either of the two institutions and diagnosed with both MMVD and cardiomegaly on echocardiography were eligible for inclusion. The inclusion period was between May 2013 and December 2015, before the EPIC study was published [4], and so before the evidence was available that dogs in Stage B2 MMVD benefitted from pimobendan medication. Therefore, some dogs were not prescribed any medication at the time of diagnosis. All patients had undergone a complete physical examination, blood pressure measurement via the Doppler method (Parks Medical Electronics Inc., Aloha, OR, USA), and Doppler echocardiography. Other investigations were performed based on clinician preference.

The pimobendan group included dogs that, at the time of diagnosis (baseline, between June 2015 and October 2018), satisfied both echocardiographic criteria outlined in the most recent ACVIM consensus [3] for stage B2 MMVD: LA/Ao ≥ 1.6 and LVIDdN ≥ 1.7. At the time of diagnosis, this population was started on chronic pimobendan monotherapy (dose range: 0.2–0.3 mg/kg PO q12h). Dogs having received or currently receiving other cardiac medications, including angiotensin-converting enzyme inhibitors (ACE-Is), spironolactone, diuretics (furosemide, torasemide, hydrochlorothiazide, amiloride), or amlodipine, were excluded. Other exclusion criteria included dogs with congenital cardiac disease, dilated cardiomyopathy, systemic hypertension (>160 mmHg), or other significant systemic disease. Baseline and follow-up data were retrieved from patient records. For dogs that received additional cardiac medications or were confirmed via radiography to have progressed into CHF, the visits up to, but not including the visit associated with these events, were selected for analysis. Inclusion of dogs being treated for CHF was outside the scope of the current study, which focussed on pimobendan monotherapy as the current recommendation for dogs with stage B2 MMVD, with additional cardiac therapies possibly confounding echocardiographic heart chamber measurements.

The control group consisted of asymptomatic dogs that were presented for investigation of a heart murmur prior to the publication of the EPIC study [4] and diagnosed with MMVD and cardiomegaly using echocardiography. These dogs had not previously received any cardiac medication, including pimobendan, as there was no supportive evidence for treatment at that time. The same echocardiographic inclusion (LA/Ao ≥ 1.6 and LVIDdN ≥ 1.7 to confirm they were stage B2) and exclusion criteria applied to the pimobendan group were applied to the control group. Control group data were only retrieved from the first post-inclusion echocardiogram in a time frame that matched the study cohort.

### 2.2. Echocardiography

Echocardiographic studies were performed by ECVIM-CA (European College of Veterinary Internal Medicine—Companion Animals) diplomates, RCVS (Royal College of Veterinary Surgeons) diplomates, or residents under direct supervision with either a Vivid 7 or Vivid 9 echocardiography machine (GE, Little Chalfont, Buckinghamshire, UK) equipped with 2–4 or 4–7 MHz cardiac phased array transducers. The identity of the operator was not retrieved for the purposes of the study, and serial echocardiograms were not always undertaken by the same operator. At least three different operators participated from institution 1, and two more participated from institution 2. A standard full echocardiographic examination was performed, including 2D, M-mode, Doppler (colour, pulsed wave, and continuous wave), and tissue Doppler imaging. Measurements and reporting were performed offline with EchoPac software v.202 (GE, Buckinghamshire, UK) by the same operator who had performed the echocardiographic examination.

The LA/Ao ratio was obtained from the right parasternal short-axis view, measured early in diastole [15]. The LVIDd was obtained from a 2D-guided M-mode view just apical to the chordae tendineae in the right parasternal short-axis view [16] and normalised to body weight using allometric scaling [5] to obtain LVIDdN. This followed the methodology described in the EPIC study [4].

### 2.3. Statistical Analysis

Statistical analyses were performed using commercially available software (Microsoft Excel v.16 Analysis Toolpak, Microsoft Office 365, Microsoft, Redmond, Washington, DC, USA and SigmaPlot v. 15, Systat Software Inc., Palo Alto, CA, USA). Data were tested for normality by visual inspection of the Q-Q plots and with the Shapiro–Wilk test. The Brown–Forsythe equal variance test was also applied. Since some data were not normally distributed, all data are presented as median and interquartile range (IQR: 25th to 75th percentiles).

The baseline characteristics of the two groups were compared. A Chi-squared test was used for sex distribution, a Fisher’s exact test for proportion of cavalier King Charles spaniels (CKCS—due to their predisposition to the disease), an unpaired *t*-test for age, a Mann–Whitney *U* test for body weight and baseline echocardiographic measurements, and a Welch *t*-test for the number of days between baseline and subsequent post-inclusion echocardiographic examinations.

A two-way repeated measures (mixed) analysis of variance (ANOVA) was used to compare echocardiographic variables between groups (control vs. pimobendan) and between baseline and subsequent post-inclusion visits (time). The data organised for this analysis passed the Shapiro–Wilk and Brown–Forsythe tests and so met model assumptions. For the treatment group, the post-inclusion visit was the first examination after initiating pimobendan, whereas for the control group, it was the first re-examination after inclusion. Post hoc multiple comparisons were performed using the Bonferroni *t*-test. If there was significant interaction between the group and baseline/post-inclusion echocardiogram (time), this was noted.

The differences at different time points among the study cohort (longitudinal analysis) were visualised by plotting the echocardiographic study data against the actual time of serial studies following inclusion (spaghetti plot). Only one additional time point was available for the control group, but a number of studies were available at various times for the pimobendan group. Given the variety of timings of serial studies for each dog, longitudinal repeated measures statistical analysis was not pursued.

For all tests, significance was accepted if the *p* value was <0.05.

## 3. Results

The control group comprised seven dogs (four males and three females). The pimobendan group consisted of 24 dogs (12 males and 12 females). For both groups, age and body weight are shown in Table 1. There were no significant differences between groups pertaining to age, sex distribution, proportion of CKCS versus other breeds, body weight, number of days between examinations, and echocardiographic values at baseline (Table 1).

The results of the two-way repeated measures ANOVA are shown in Table 2, with a graphical representation of LVIDdN and LA/Ao data points as box and whisker plots (Figure 1a and Figure 1b, respectively). For LVIDdN, there was not a significant group effect (*p* = 0.810), but there was a significant interaction between group and time (*p* = 0.040), with a significant reduction in LVIDdN occurring over time in the pimobendan group (*p* = 0.038). The LA/Ao decreased over time in both groups (*p* = 0.010) and to similar extents (*p* = 0.561, interaction *p* = 0.855).

The ACVIM consensus requires dogs to have both an LA/Ao > 1.6 and an LVIDdN > 1.7 to be classified as stage B2 MMVD [3]. Following pimobendan monotherapy, 10 of the 24 treatment dogs (42%) no longer met the ACVIM criteria for stage B2 MMVD at the first follow-up. This group comprised three dogs (13%) with an LA/Ao ratio of less than 1.6, four dogs (17%) with an LVIDdN of less than 1.7, and three dogs (13%) with both values under the cut-off for MMVD stage B2.

When the echocardiographic variables were retrieved for longitudinal analyses, including the subsequent examinations in the pimobendan group, there were only 10 out of the 24 dogs that had studies carried out at each time point—baseline, <90 days, and <180 days after treatment initiation—so serial longitudinal statistical evaluation was not performed. The trends in LVIDdN over time from all 24 patients in the pimobendan group and all seven dogs in the control group are shown in the Appendix A (Appendix A, respectively). The complete dataset from the pimobendan and the control groups are given in Appendix A. 

## 4. Discussion

This study supported a treatment effect of monotherapy pimobendan with a significant reduction in diastolic left ventricular size over time compared with the control dog population. There is previous evidence supporting our finding, with documented reverse remodelling following the administration of pimobendan in Dobermans with dilated cardiomyopathy [13] and dogs with MMVD and CHF [8,9]. However, our results suggest this same phenomenon may occur in dogs with stage B2 MMVD, i.e., prior to the development of CHF. Our findings build on the evidence provided by the longitudinal data analysis from the EPIC study in which LVIDdN and LA/Ao ratio were decreased in preclinical MMVD dogs 35 days after starting pimobendan [12].

As a result of the observed decrease in both LVIDdN and LA/Ao ratio in our study, ten of the 24 dogs that received pimobendan no longer met the echocardiographic classification criteria for stage B2 MMVD and instead could have been classified as “stage B1”. The ACVIM staging scheme [3] does not consider the possibility that dogs with MMVD can revert to a previous stage of disease. Thus, we suggest that animals with a documented reduction in the left heart chambers after pimobendan treatment that no longer meet the criteria for stage B2 disease should instead be described as stage B2 reverse remodelled. The clinical significance of the reclassification we propose is that such dogs administered pimobendan prior to MMVD staging could be misclassified at a subsequent first echocardiographic examination as stage B1 and have pimobendan withdrawn to their detriment when, in fact, they are stage B2 reverse remodelled and have benefitted from pimobendan intervention. The distinction between stage B1 and stage B2 reverse remodelled could only be safely made if results from baseline staging are available. This reinforces the importance of MVMD staging prior to initiating pimobendan therapy, as per the consensus [3].

The LA/Ao ratio was slightly smaller in the control group as well as in the post-inclusion examination in pimobendan-treated dogs. It cannot be definitively stated that the reduction in the left atrial size in the treatment group was due to pimobendan alone. The relatively low reproducibility and repeatability of the LA/Ao ratio [17] may have influenced the measurements at different time points and masked a potentially mild therapeutic effect. Previous publications have similarly highlighted the intra and inter-observer variability when measuring the LA/Ao ratio [15,18,19,20,21]. This is particularly true when the increase in size is mild [21] and within the range expected of many stage B2 MMVD dogs. Other 2D left atrial measurements are more reproducible and sensitive [22]; however, the LA/Ao ratio was used in this study in accordance with the current consensus (and according to the EPIC study) and because the retrospective nature of the study limited our ability to select alternative approaches. The reason for the control group tending to have a smaller LA/Ao ratio on the post-inclusion echocardiogram cannot be easily explained, and it is inconsistent with the pathophysiology and expected progression of the disease. Interestingly, a similar finding was observed in the EPIC study, which described a reduction in the LA/Ao ratio in dogs receiving a placebo [4]. It was thought to be a consequence of operator-related overestimation of the left atrial size when classifying animals as stage B2 to prescribe medication. Then, the values at re-examination, when over-estimation is less likely to happen in a treatment group patient, would be closer to the true population mean (a phenomenon called ‘regression to the mean’ [23]). However, this phenomenon is less likely in this study because the dogs were classified prior to the publication of the EPIC trial. It is more likely a result of the small number of control dogs or that the control dogs may have been at an earlier stage of stage B2 with variations in inter and intra-observer measurements.

It is well-accepted that increased cardiac size is associated with an increased risk of developing CHF [24,25,26]. The documented reverse remodelling in the EPIC study [4] suggested a decrease in cardiac size as a mechanism for the prolongation of the preclinical phase of MMVD in dogs receiving pimobendan. Our findings give weight to this mechanism, with similar changes documented. It is proposed that pimobendan’s combination of selective phosphodiesterase 3 inhibition and calcium sensitization properties (‘inodilater’) favours forward stroke volume and, therefore, reduces mitral regurgitant fraction of the total left ventricular systolic stroke volume. This decrease in left-sided volume overload may result in reduced left atrial and/or left ventricular size. Furthermore, the positive inotropic effect of pimobendan will reduce left ventricular systolic diameter and volume, reducing mitral annular stretch and thereby reducing secondary mitral regurgitation. These haemodynamic improvements may reduce cardiac size. The enhanced forward stroke volume may improve renal perfusion and down-regulate the renin–angiotensin–aldosterone system, reducing sodium and water retention and ultimately reducing preload and volume overload [27]. Together, these changes may delay the time before the onset of congestive heart failure. In addition, recent research also suggests that pimobendan may play a role in preserving mitochondrial function and myocyte ultrastructure in a rat model [28]. Previous studies that analysed longitudinal radiographic data in untreated dogs with preclinical MMVD showed a linear increase in heart size followed by an exponential increase as dogs were closer to developing CHF [29,30]. In the current study, dogs with MMVD stage B2 treated with pimobendan showed a non-linear progression. As Appendix A shows, some dogs receiving pimobendan tend to decrease their LVIDdN for a variable time, and then it increases again (i.e., a U-bend), but other dogs do not show this response. This variability in time of repeat echocardiography and the variable response to the pimobendan in individual dogs may explain the lack of significant group effect overall in our two-way repeated measures ANOVA. The small group sizes, especially in the control group may also have resulted in statistical underpowering.

There are several limitations that must be considered in this retrospective study. Firstly, the sample size of the control group was small. It was difficult to find cases that had not received pimobendan or any other medication even prior to the publication of the EPIC study. The longitudinal analyses were also challenging because the time between reassessments was not standardised. However, the main limitation of this study is the lack of intra- or inter-operator variability studies with multiple echocardiographers performing the studies and providing measurements (at least five in two institutions). Echocardiographic measurement variability has been reported in several studies, and we have reported a reduction in LVIDdN that lies within the previously published coefficient of variation [17,31,32]. However, the opposing trend in the control group, within which the same repeatability would apply, may suggest a treatment effect from pimobendan rather than normal variability in measurements. A prospective study involving a larger study population and far fewer echocardiographers should be undertaken to confirm the current findings. However, withholding pimobendan in dogs with stage B2 MMVD would now be considered unethical, given the evidence of significant prolongation of the preclinical period prior to the onset of CHF. Finally, the echocardiographers were not blinded; however, as the generation of data preceded the conception of this study, bias during the echocardiographic acquisition of measurement was unlikely.

## 5. Conclusions

In conclusion, pimobendan monotherapy reduces LVIDdN and possibly LA/Ao in dogs with MMVD stage B2, with the possibility of misclassification of the disease stage if started prior to initial echocardiography. Therefore, it is important that MMVD staging is performed prior to initiating pimobendan to avoid mistaking reverse remodelled stage B2 as stage B1, complicating treatment plans.

## Figures and Tables

**Figure 1 vetsci-11-00594-f001:**
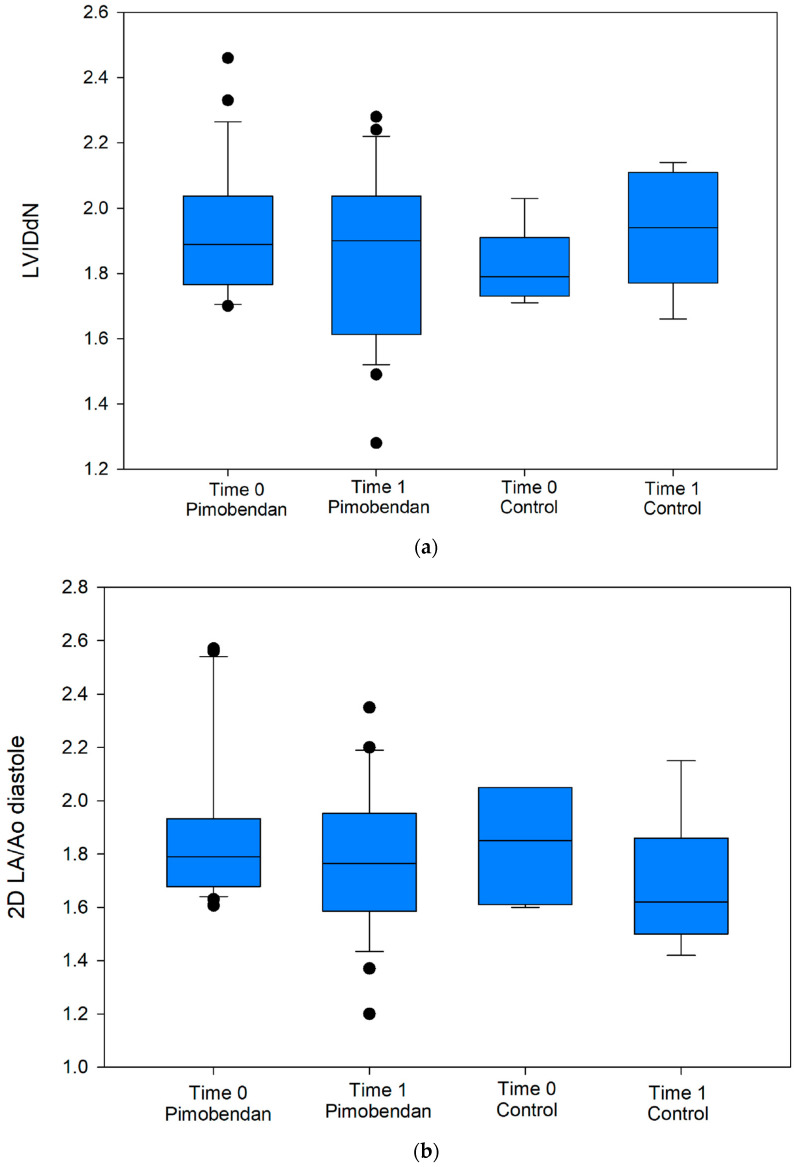
(**a**) Box and whisker plot demonstrating the distribution of LVIDdN across the pimobendan group and control group at the time of enrollment (Time 0) and at first recheck (Time 1). Boxes represent the interquartile range, with the median indicated with the line dividing the box. Outliers are represented by solid dots. (**b**) Box and whisker plot demonstrating the distribution of LA/Ao across the pimobendan group and control group at the time of enrollment (Time 0) and at first recheck (Time 1). Boxes represent the interquartile range, with the median indicated with the line dividing the box. Outliers are represented by solid dots.

**Table 1 vetsci-11-00594-t001:** Comparative demographic and baseline echocardiographic characteristics of the control and study cohorts. Data expressed as median (IQR: 25th–75th percentile).

	Controls (n = 7)	Pimobendan (n = 24)	*p*-Value
Sex	Males:4 Females: 3	Males: 12 Females: 12	0.923
Breed	CKCS: 3 Other breeds: 4	CKCS: 12 Other breeds: 12	0.923 (CKCS vs. non-CKCS)
Weight (kg)	8.0 (7.3–9.4)	9.0 (7–10)	0.388
Age (years)	7 (7–9)	10 (8–11)	0.125
LA/Ao	1.85 (1.61–2.05)	1.79 (1.68–1.93)	0.684
LVIDdN	1.79 (1.73–1.91)	1.89 (1.77–2.04)	0.156
Time between exams (days)	182 (88–242)	95 (72–148)	0.139

**Table 2 vetsci-11-00594-t002:** Two-way repeated measures analysis of variance (ANOVA) for group (pimobendan/control) and time (baseline/post-inclusion echocardiography). Data are presented as median (IQR: 25th to 75th percentile).

Variable	Baseline Control Group (n = 7) (Median, IQR)	Post-Inclusion Control Group (n = 7) (Median, IQR)	Baseline Pimobendan Group (n = 24) (Median, IQR)	Post-Inclusion Pimobendan Group (n = 24) (Median, IQR	Overall *p* Value	Multiple Comparison Baseline and Post-Inclusion Control Group (Over Time)	Multiple Comparison Baseline and Post-Inclusion Pimobendan Group (Over Time)	Interaction Between Group and Time
LA/Ao	1.85 (1.61–2.05)	1.62 (1.50–1.86)	1.79 (1.68–1.93)	1.77 (1.59–1.95)	group: *p* = 0.561 time: *p* = 0.010	-	-	No (*p* = 0.855)
LVIDdN	1.79 (1.73–1.91)	1.94 (1.77–2.11)	1.89 (1.77–2.04)	1.90 (1.61–2.04)	group: *p* = 0.810 time: *p* = 0.939	*p* = 0.216	*p* = 0.038	Yes. (*p* = 0.040)

## Data Availability

The original contributions presented in this study are included in the article/Appendix A. Further inquiries can be directed to the corresponding authors.

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
