# Peer review of "Echocardiographic Changes in Dogs with Stage B2 Myxomatous Mitral Valve Disease Treated with Pimobendan Monotherapy"

_vetsci, 2024, doi:10.3390/vetsci11120594_

Round 1
Reviewer 1 Report
Comments and Suggestions for Authors
Review of “Echocardiographic changes in dogs with stage B2 myxomatous mitral valve disease treated with pimobendane monotherapy”
The authors present a retrospective study of the effects of pimobendan monotherapy on the progression of echocardiographic signs of myxomatous mitral valve disease. Generally speaking, this is a useful study that adds to our knowledge of the topic; however, I think it would benefit from some changes and additions, which I will outline below:
Simple Summary
I’m not entirely sure whom the simple summary is aimed at for the purposes of this journal, but if it is intended for the general public, the language used is far too technical. You need to define B2 in simple terms, and also consider changes such as “degenerative heart valve disease” instead of “myxomatous mitral valve disease”, “ultrasound” instead of “echocardiography”, “pimobendan only” as opposed to “monotherapy”, “heart size” as opposed to “cardiac size”, and so on.
Abstract
I’m unclear on what the authors mean by “there was no significant group effect but LVINdN significantly reduced over time in the pimobendan group”, please clarify because this sounds self-contradictory at first sight.
Introduction
Line 49, add a snippet on breed and size predispositions, which is particularly relevant given your patient demographics. Other than this detail the section is solid and well-written.
M&M
Line 117, why did you exclude patients with systemic hypertension?
Based on the authors’ description of their data, I also think a multiple regression model would be beneficial to employ here.
As a general comment, I see no indication that the authors considered patient age as a variable affecting outcome, which I think is a significant oversight. Please include this variable and see what the effect is.
Results
Table 1, While the authors found no age difference between their population, I think it is still worthwhile to look at age as a variable determining outcome, as above. Age is the single greatest risk factor for a plethora of diseases including MVD, and it is reasonable to think it will affect prognosis and progression over time.
Figure 1 is not very visually appealing, I think it would be better presented using boxplots instead of individual data points. Either way, these are not histograms.
Figure 2, while I commend the authors’ intention of presenting complete data, I think these two plots are probably best included as supplemental materials.
Discussion
Line 241, I think these studies should probably be mentioned in the introduction instead.
Line 278, I’m not sure the argument that reduced cardiac size counts as a mechanism of delayed onset of CHF is sound – rather, I’d consider those effects to be two sides of the same figurative coin. Please discuss in more detail why you think one is causative of the other and how the data support that claim.
Line 299, I think you’re selling yourselves short here by burying this part in the middle of the discussion. Being able to reverse dogs from B2 back to B1 is a significant positive finding and you should give it more weight in the overall presentation of the paper.
Author Response
Thank you very much for your constructive comments on our manuscript. We have updated the manuscript and responded to your comments - please see the attached file.

Reviewer 2 Report
Comments and Suggestions for Authors
The presented project is interesting; However, the article is rather challenging to understand. It was only upon reading page 4 that I began to understand the study. It involves a retrospective analysis of two groups of dogs: one diagnosed as B2 and treated with pimobendan, and another group for which classification is unspecified. Based on the measurements in Table 1, I assume these dogs would also fall under the B2 classification, although this is not explicitly indicated. These latter dogs did not receive pimobendan, as the EPIC study had not yet been published at the time of this study.
**Lines 23-29 and 39-43**: The suggestions presented are problematic as they are based solely on the study's findings and fail to account for the study's limitations; statements like these should be avoided.
**Line 35**: Duration? Please specify.
**Keywords**: Reword, as the terms are generally generic.
**Line 56**: Consider replacing "preclinical" with "prior to CHF."
**Line 75**: Consider changing "sum" to "score."
**Lines 88-90**: Given the study's limitations, relying only on echocardiography and the limited measurements performed, assessing cardiac remodeling is overly ambitious.
**Line 98**: Statistical analyses, such as a correlation coefficient between classes, could validate the differences in determinations across operators and institutions.
**Lines 105-106**: The terms "data" and "impact" add little to the study; consider revising or omitting this sentence.
**Lines 118-120**: How might including these animals negatively impact the study? Please add insights on this matter.
**Line 124**: Was diagnosis determined via echocardiogram? Were these cases indeed B2? Clearer specification is needed, as this section is difficult to follow. Were they asymptomatic yet presented values ​​as shown in Table 1? No mention is made of lethargy, exercise intolerance, syncope, or related conditions, such as cough or dyspnea.
**Line 138**: Why was the intraclass correlation coefficient not calculated?
**Lines 140-141**: Although various echocardiographic modes were used, only atrium and ventricle data are included, leading to the exclusion of valuable information, such as diastolic flow pressures, TDI, regurgitation degree, etc.
**Table 1**: There is a large discrepancy in timing between check-ups, and they lack uniformity across groups.
**Lines 198-199**: How can these findings be explained in the group not receiving pimobendan? Did LVIDn increase in those receiving pimobendan?
**Lines 204-207**: Requires clearer explanation.
**Line 251**: This statement appears inaccurate, or Table 1 may be incorrect.
**Line 255**: Given this, evaluating correlation coefficients and intra- and inter-operator effects is crucial. Additional measurements, such as the AI:AO diameter, area, pressures, TDI, and regurgitation, would aid in determining why a greater or lesser diameter was observed.
**Lines 303-309**: Commenting on these statements or hypotheses with such a small sample size and study limitations seems ambitious. Rewording is recommended.
**Figures 1A and 1B should be reformatted for improved visualization.
Author Response
Thank you very much for your detailed comments. We hope we have addressed these and revised the manuscript accordingly. Please see our responses in the attached document.

Round 2
Reviewer 1 Report
Comments and Suggestions for Authors
Thank you for addressing my comments, the manuscript is acceptable for publication in this form.